# Cytokine Profiling of End Stage Cancer Patients Treated with Immunotherapy

**DOI:** 10.3390/vaccines9030235

**Published:** 2021-03-08

**Authors:** Marco Carlo Merlano, Andrea Abbona, Matteo Paccagnella, Antonella Falletta, Cristina Granetto, Vincenzo Ricci, Elena Fea, Nerina Denaro, Fiorella Ruatta, Anna Merlotti, Oscar Bertetto, Nicola Crosetto, Danilo Galizia, Marco Basiricò, Loretta Gammaitoni, Dario Sangiolo, Massimo Aglietta, Ornella Garrone

**Affiliations:** 1Experimental Cell Therapy Lab, Department of Medical Oncology, Candiolo Cancer Institute, FPO-IRCCS, 10060 Turin, Italy; loretta.gammaitoni@ircc.it (L.G.); dario.sangiolo@ircc.it (D.S.); massimo.aglietta@unito.it (M.A.); 2Translational Oncology, ARCO Foundation, 12100 Cuneo, Italy; abbona.andrea@gmail.com (A.A.); matteo.babeuf@gmail.com (M.P.); antonella.falletta.92@gmail.com (A.F.); ornella.garrone@gmail.com (O.G.); 3Department of Medical Oncology, S. Croce e Carle Teaching Hospital, 12100 Cuneo, Italy; granetto.c@ospedale.cuneo.it (C.G.); vincenzoricci22@libero.it (V.R.); fea.e@ospedale.cuneo.it (E.F.); nerinadenaro@gmail.com (N.D.); fiorella.ruatta@gmail.com (F.R.); 4Department of Radiotherapy, S. Croce e Carle Teaching Hospital, 12100 Cuneo, Italy; merlotti.a@ospedale.cuneo.it; 5Rete Oncologica del Piemonte e Della Valle d’Aosta, 10125 Turin, Italy; obertetto@cittadellasalute.to.it; 6Department of Medical Biochemistry and Biophysiscs, Karolinska Institute, 17177 Stockholm, Sweden; nicola.crosetto@scilifelab.se; 7Multidisciplinary Oncology Outpatient Clinic, Candiolo Cancer Institute, FPO-IRCCS, 10060 Turin, Italy; danilo.galizia@ircc.it; 8Department of Oncology, University of Turin, 10043 Turin, Italy; marco.basirico@ircc.it; 9Breast Unit, S. Croce e Carle Teaching Hospital, 12100 Cuneo, Italy

**Keywords:** end-stage cancer patients, immunotherapy, cytokinome

## Abstract

Published data suggest that immunotherapy plays a role even in patients with very advanced tumours. We investigated the immune profile of end-stage cancer patients treated with immunotherapy to identify changes induced by treatment. Breast, colon, renal and prostate cancer patients were eligible. Treatment consisted of metronomic cyclophosphamide, low-dose interleukin-2 (IL-2) and a single radiation shot. A panel of 16 cytokines was assessed using automated ELISA before treatment (T0), after radiation (RT; T1), at cycle 2 (T2) and at disease progression (TPD). Receiving operating characteristic (ROC) analysis was used to identify cytokine cut-off related to overall survival (OS). Principal component analysis (PCA) was used to identify the immune profile correlating better with OS and progression-free survival. Twenty-three patients were enrolled. High IL-2, low IL-8 and CCL-2 correlated with OS. The PCA identified a cluster of patients, with high IL-2, IL-12 and IFN-γ levels at T0 having longer PFS and OS. In all cohorts, IL-2 and IL-5 increased from T0 to T2; a higher CCL-4 level compared to T2 and a higher IL-8 level compared to T0 were found at TPD. The progressive increase of the IL-10 level during treatment negatively correlated with OS. Our data suggested that baseline cytokine levels may predict patients’ outcome and that the treatment may affect their kinetic even in end-stage patients. Cytokine profiling of end-stage patients might offer a tool for medical decisions (EUDRACT: 2016-000578-39).

## 1. Introduction

During the past decade, the development of immunotherapy (IO) has led to a dramatic improvement in the treatment of solid tumours. Indeed, cytotoxic T-lymphocyte antigen-4 (CTLA-4) and programmed death 1 (PD-1)/programmed death-ligand 1 (PD-L1) axis inhibitors have produced spectacular results in cancers on which historically conventional anticancer therapies have limited impact, including malignant melanoma and lung cancer. Following these early successes, many other immune agents were developed and are under clinical evaluation, either as single agents [1] or in combination with other immunotherapies, radiotherapy [2] or chemotherapy [3]. This huge effort is also due to the knowledge that there are many immune escape mechanisms, frequently redundant, which explains why many patients do not benefit from IO, as recently reviewed [4,5].

IO has shown various levels of activity have been observed in most solid tumours, and this observation supports the hypothesis that the activity of IO is at least in part unrelated to the tumour type and is due to the modulation of the immune system rather than to a direct antitumour activity.

Remarkably, IO can provide benefits to patients with advanced disease stages [6,7], which has led to the introduction of the term “Lazarus effect” [8].

Therefore, there is a growing debate on whether even patients in the preterminal phase should be treated with IO [9]. Unfortunately, data supporting the role of this treatment modality in these patients remain very scarce.

To explore the activity of IO in preterminally ill cancer patients, in 2016, we designed a translational exploratory study in end-stage patients suffering from colon, breast, renal and prostate cancers treated with metronomic cyclophosphamide (CTX), low dose interleukin-2 (IL-2) and radiation (RT).

Our goal was to generate data on the effect of a combined immune approach on the immune profile of end-stage patients, by the analysis of circulating immune cells, IL and cytokines of end-stage patients and correlate them with the outcome. Our ambition was to identify changes or baseline values of specific cytokines able to predict clinical benefit of the treatment and, in the future, to give clinicians tools to drive clinical decisions toward new therapies or best supportive care in the end stage. Here, we report the results of this study, focusing on the analysis of 16 circulating cytokines. The cytokines were chosen, because they are important drivers of the tumor microenvironment toward either the Th1 (IL-2, IFN-γ, TNF-α, IL-12, IL-15 and CXCL-10) or the Th2 (CCL-2, CCL-4, CCL-22, IL-4, IL-5, IL-6, IL-8, IL-10, IL-13 and TGF-β) profile [10]. At the time of this communication, the analysis of circulating immune cells is still ongoing.

## 2. Materials and Methods

### 2.1. Patient Population

Patients were enrolled, if they were considered as having end-stage disease. The definition of “end-stage disease” identifies patients with multiple metastases previously treated with all the available approved lines of therapy and lacking further adequate treatments according to their disease and in line with the Italian National Guidelines [11]. Patients with histologically confirmed metastatic breast, colon, kidney or prostate cancer were considered eligible for this study. All patients needed to have a life expectancy of at least 3 months or more, Eastern Cooperative Oncology Group Performance Status (ECOG P.S.) of 2 or less, adequate cardiac, liver and hematological functions. Measurable diseases according to Response Evaluation Criteria in Solid Tumors (RECIST) criteria version 1.1 [12] and at least one deposit susceptible of radiotherapy were among the inclusion criteria. Patients with immunological disorders, active acute or chronic infections, brain metastases, steroids treatment and any other clinical, familiar or social conditions that could preclude the adherence to the protocol in the judgment of physicians were excluded from the study. The study was approved by the local ethical committee (ONCO 2016: 264/265-268/269), and all patients signed informed consent.

### 2.2. Treatment

The first course of treatment (28 days) consisted of 50 mg CTX orally once a day continuously and 500,000 International Unit (IU) IL-2 twice a day subcutaneously, every other week, starting 8 days after the beginning of CTX. A single 8 Gy flash of RT was delivered to one single lesion, seven days after the beginning of CTX. The following courses were repeated every 28 days, without RT. The treatment continued until disease progression (TPD), unacceptable toxicity or patients’ refusal. The combined treatment was chosen due to the selective downregulation of CD4+, CD25+ and FoxP3+ T cells (Tregs) by metronomic low-dose CTX [13], the expansion effect of IL-2 on T cells and the immunogenic cell death (ICD) induced by RT, supposed to act as a sort of “self-vaccination”. Patients were treated at the Medical Oncology Department, S. Croce e Carle Teaching Hospital, Cuneo, Italy and followed up until death.

### 2.3. Sample Collection and Handling

Fifteen millilitres (mL) of peripheral blood were collected in three vacutainers at different time points: baseline (T0), the day after RT (T1) and day 1 of cycle 2 (T2) and at TPD. Plasma samples were obtained by centrifugating one vacutainer at 1500 g/m for 10 min. Aliquots of 600 μL in 2 mL cryovials were kept at −80 °C. Peripheral blood-mononucleated cells (PBMCs) were obtained through Ficoll separation, following standard procedures. The cells were cryo-preserved using a Planer instrument (Kryo 560-16) or by slow freezing (2 h at 4 °C, 2h at −20 °C, then −80 °C) in aliquots of 3 × 10^6^ cells. All cryovials, both plasma and PBMCs, were kept at −80 °C. Circulating cytokines/IL analyses were performed at the ARCO Foundation Lab, S. Croce e Carle Teaching Hospital, Cuneo, Italy. Circulating PBMCs analyses are on-going at the Medical Oncology Experimental Cell Therapy Lab, Candiolo Cancer Institute, FPO-IRCCS, Candiolo (Turin), Italy.

### 2.4. Cytokines Analyses

A panel of 16 cytokines (CCL-2, CCL-4, CCL-22, CXCL-10, IFN-γ, IL-2, IL-4, IL-5, IL-6, IL-8, IL-10, IL-12, IL-13, IL-15, TGF-β and TNF-α) was assessed using the Simple Plex system (ProteinSimple, San Jose, CA, USA). Ella Simple Plex System is a microfluidic device which uses prekitted immunoassay cartridge. The sample ran through a microfluidic channel that bound the protein of interest. Briefly, a two-fold dilution of each plasma sample was spun for 5 min at 1000× *g* and added to the Simple Plex cartridge for each cytokine, except TGF-β, which was previously activated (1 N HCl, and then neutralized with 1.2 N NaOH/0.5 M HEPES) to a final dilution with a volume ratio of 1:15. The cartridge was inserted into the Ella reactor and run for 90 min. The concentrations were expressed in pg/mL. All the samples were assayed in duplicate.

### 2.5. Statistical Analysis

The basal values of circulating cytokines and chemokines were compared with the outcome to establish their prognostic values. The values of circulating cytokines and chemokines at different time points were compared to identify the effect of treatment. The correlation between the values of circulating cytokines at different time points and objective responses/outcomes was essentially descriptive. No a priori sample size calculation and statistical power were performed due to the exploratory nature of the study. The longitudinal analysis of cytokines distribution at each time point was performed by the nonparametric Mann–Whitney U test, in GraphPad PRISM V.5. Baseline cytokines values were used to find prognostic markers for overall survival (OS). The patient population was divided in two groups depending on the T0 values of each cytokine using cut-off values calculated with the receiving operating characteristic (ROC) curve analysis. The cut-off was defined as the point on the ROC curve with the best compromise between sensitivity and specificity. Principal component analysis (PCA) was used to reduce the complexity of high dimensional data, visualize them and overcome multicollinearity of variables. PCA analysis was performed with R software (version 3.5.3 “great Truth”). The T0 values of the previous defined panel of cytokines were used as variables. Progression-free survival (PFS) and OS were estimated by the Kaplan–Meier method and were compared using the log-rank test. Hazard ratios (HRs) were calculated using the cox proportional hazards model using SPSS 24.0 (IBM Corporation, Armonk, NY, USA). Considering the large number of comparisons planned, we adjusted for multiple hypothesis testing using the Bonferroni correction method [14].

Responses were evaluated according to the RECIST criteria [12], and clinical benefit (CB) was defined as a stable disease lasting 24 weeks or more.

PFS was defined as the time elapsed between the date of the treatment initiation and the date of the TPD or death, whichever occurred first. OS was defined as the interval between the date of the treatment start and the date of the death for any causes.

## 3. Results

From September 2016 to May 2019, 23 patients with end-stage tumours (6 patients with breast cancer (TB), 4 patients with prostate cancer (TP), 4 patients with kidney cancer (TK) and 9 patients with colon cancer (TC)) were enrolled. Patients’ characteristics, including the previous line of treatment, time to treatment failure, PFS, OS, basal lactate dehydrogenase (LDH) value, neutrophil/lymphocyte ratio (NLR) and the corresponding code number of each patient in the PCA figures, are reported in Table 1. The response to the last previous two lines of therapy for each patient is also reported. Overall, the patients received a median of 4 previous lines of treatment (range: 2–9). We did not observe any objective response. Five patients (22%) showed a stable disease (SD) after the first evaluation. Only 1 of them (4%) obtained a CB. The overall median PFS was 2.9 months (95% confidence interval (CI): 2.6–3.2), and the median OS was 6.6 months (95% CI: 2.8–10.3). The median PFS and OS by the primary tumour site are reported in Table 2.

We did not find any relationship between the basal level of LDH or NLR and OS in ROC analysis.

Five patients received further treatments at PD. Three had two treatment lines (see Table 1: TC1, TC6 and TB3) and two had one line (TC4 and TP1). The best response was SD, achieved in pts TC1, TP1 and TB3. Toxicity was evaluated according to the National Cancer Institute Common Terminology Criteria for Adverse Events (NCI-CTCAE, version 4). Treatment was extremely well tolerated, without any grade III or IV toxicity. Most common grades I and II adverse events were anaemia in 10 patients. Fatigue and nausea were observed in nine patients each, and thrombocytopenia was observed in one patient.

We used T0 cytokines levels to find prognostic markers for OS. The patients’ population was divided into two groups (above or below the median OS) and the T0 values of the cut-off points of cytokines were calculated with ROC analysis. Among all the cytokines, we were able to identify cut-off values for IL-2 (0.055 pg/mL; Area Under the Curve (AUC): 0.193; 95% CI: 0.009–0.377; *p* = 0.013), IL-8 (10.75 pg/mL; AUC: 0.860; 95% CI: 0.701–1.000; *p* = 0.003) and CCL-2 (207.5 pg/mL; AUC: 0.848; 95% CI: 0.693–1.000; *p* = 0.005). Using them, we observed a significant improvement in OS in patients with T0 levels of IL-2 higher than its cut-off values and with T0 values of IL-8 and CCL-2 lower than their cut-off values (HR: 0.204; *p* = 0.002; HR: 0.096; *p* = 0.001; HR: 0.338, *p* = 0.031) (Figure 1). 

A summary of the scientific meanings of these results is shown (Figure 2).

We then clustered the patient population in subgroups depending on their T0 levels of plasma cytokines using the PCA of 15 variables, normalised by z-scores (T0 values of each cytokine/chemokine of our panel). The two principal components (PC1 and PC2) explained 22.7% and 13.8% of the data variance, respectively. Subsequently, we used two coincident hyperplanes x-axis and y-axis to subdivide the plane in four areas. Higher values of CCL-2, IL-6 and IL-8 contributed to area 1; the values of CCL-4, IL-4, IL-10, IL-13, IL-15, TNF-α and TGF-β contributed to area 2; the values of CCL-22, IFN-γ, IL-2, IL-5 and IL-12 contributed to area 3; no specific cytokines were found to contribute to area 4 (Figure 3).

We clustered the patients according to their distributions in the four areas and calculated their PFS and OS values. Four patients were allocated in area 1 (1B, 1K, 2C) with a median PFS value of 2.1 months (95% CI: 1.2–3.0) and a median OS value of 2.9 months (95% CI: 0.41–5.4); Area 2 included five patients (2B, 1P, 2C) with a median PFS value and a median OS value of 2.8 months (95% CI: 2.5–3.0) and 5.0 months (95% CI: 1.2–8.7), respectively; area 3 contained seven patients (1B, 2P, 3K, 1C) with a median PFS value of 6.6 months (95% CI: 0.0–14.0) and a median OS value of 13.8 months (95% CI: 6.8–20.8); area 4 had also seven patients (2B, 1P, 4C) with a median PFS value of 2.8 months (95% CI: 2.7–2.9) and a median OS value of 5.6 months (95% CI: 1.3–9.9) (Figure 4a,b). The patients of area 3 had significantly longer PFS and OS values compared to the patients in areas 1, 2 and 4 cumulatively (*p* = 0.004, HR = 0.158, 95% CI = 0.044–0.563; *p* = 0.023, HR = 0.304, 95% CI = 0.109–0.849) (Figure 5a,b). However, considering also the number of previous lines of therapy and the cancer type in a multivariate analysis, area 3 was the only significant factor for PFS (*p* = 0.0449), while the number of previously treatment lines was the only factor significantly affecting OS (*p* = 0.005) (Table 3).

The patients of area 3 had significantly longer PFS and OS values compared to the patients of areas 1, 2 and 4 cumulatively (*p* = 0.004, HR = 0.158, 95% CI = 0.044–0.563; *p* = 0.023, HR = 0.304, 95% CI = 0.109–0.849) (Figure 4a,b).

We then analysed the differences in the cytokine levels at T0 for patients clustered among the areas. Briefly, we found that pts in area 4 had significantly lower IL-4 and IL-15 levels compared to those in area 2 and significantly lower IL-2, IL-10, IL-12, IL15 and IFN-γ levels compared to those in area 3. The pts in area 3 had a significantly higher IL-12 level and a significantly lower CCL-2 level compared to those in area 1. We did not find any difference at T0 among the remaining cytokines between the four clusters of patients (Figure 6).

Next, we performed the longitudinal analysis of the whole population. The IL-2 and IL-5 levels increased during treatment from T0 to T2 (*p* < 0.0001 and *p* = 0.005, respectively). The IL-2 level also significantly increased between T1 and T2 (*p* = 0.002). Considering the values at TPD, CCL-4 increased between T2 and TPD (*p* = 0.008), and the IL-8 level increased between T0 and TPD (*p* = 0.006) (Figure 7).

Considering the PCA areas (Figure 3), we clustered patients in two groups: Group A, comprising the patients in areas 1, 2 and 4 that showed a similar outcome as shown in Figure 4a,b and group B, represented only by the patients in area 3. No major differences between groups A and B were observed among the three time points, although IFN-γ, IL-2, IL-12 and CXCL-10 seemed to be generally higher in group B (Figure 8).

Considering the whole population, only the increase of IL-10 over time correlated with a significantly poorer survival (Figure 9).

A summary of the scientific meanings of the modification of IL-10 is illustrated (Figure 10).

## 4. Discussion

In this exploratory study of IO in end-stage patients, we have not observed any objective response among the 23 patients enrolled, although three of them survived more than 100 weeks from the beginning of treatment (pts TC1, TK2 and TB3). These long survivals seemed unrelated to the primary sites, number of previous treatment lines, and NLR. However, two of these patients were allocated in a favourable group, based on our PCA of cytokine kinetics following IO. We did not observe any “Lazarus effect” even if patients with kidney cancer were included, and IL-2 is a recognized treatment for this disease [15].

We have used baseline cytokines values to find prognostic markers for OS. Levels of IL-2 higher than 0.055 pg/mL (the cut-off value) were associated with good prognosis. For the best of our knowledge, the relationship between values of IL-2 and prognosis has not yet been reported. This may be due to the low levels of circulating IL-2. However, a correlation between this cytokine and prognosis may have sense, since IL-2 exerts well-known positive effects on the immune response, notwithstanding that it is also able to expand Treg cells. In our study, the Treg expansion should have been counteracted by low-dose metronomic CTX. The ongoing analysis of circulating immune cells will clarify this point.

Low levels of IL-8 and CCL-2 (below 10.75 pg/mL and 207.5 pg/mL, respectively) were also associated with longer OS. It was established that high baseline values of CCL-2 are indicators of poor PFS and OS in many solid tumours [16].

Similarly, IL-8 high plasma levels were widely associated with poor OS [17,18]. Looking at the longitudinal analysis, only the progressive increase of IL-10 at each time point correlated with a significant reduction of survival, reinforcing the negative role of this IL [19].

To take into account possible interactions, we also considered all cytokines together by clustering patients using all their T0 cytokines values in a PCA; PCA has already been used to distinguish metastatic breast cancer patients from healthy volunteers [20]. We applied the same approach to all the different types of tumour accrued in our study. We identified a group of patients, located in area 3 of a PCA biplot, with the best outcome in terms of PFS and OS with respect to the others. It is worth noting that PFS depends only on the areas of the PCA but not on the tumour types or the number of previous lines of treatment (more or less than four), while the number of previous lines of treatment is the only factor affecting OS, as shown in our multivariate analysis. Since PFS is mainly affected by the treatment, area 3 could identify a subpopulation of patients who might benefit from further treatment. The favourable behaviour observed in the patients in area 3 could be explained by the cytokine pattern driving this cluster. Indeed, the Th1 cytokines IL-2, IL-12 and IFN-γ mainly contribute to the definition of area 3. The biological roles of these cytokines are interconnected with each other. For instance, both IL-2 and IL-12 can stimulate CD3-activated T cells proliferation. Moreover, it was demonstrated that IL-12 can induce the expression of IFN-γ [21]. In line with these observations, several preclinical studies have underscored the promise of IL-12/IL-2 combination therapies [22,23]. Indeed, the two ILs synergize with each other increasing the magnitude of T cell functional responses [24].

Area 4 was characterized by the absence of driving cytokines, a clue of a silenced and compromised immunological status that could explain the poor prognosis of the patients in this area. Areas 1 and 2 were significantly influenced by Th-2-related cytokines. Interestingly, areas 1, 2 and 4 show similar behaviours in terms of PFS and OS. We speculate that a Th-2-dominant immune profile and a silenced profile similarly contributed to the same poor prognosis in these patients.

We investigated the longitudinal changes of circulating cytokines during treatment period and at TPD. We observed that, in all patients, treatment influenced the behaviour of some cytokine levels (IL-2, IL-5 and CCL-4) although this effect disappeared at TPD. We found that IL-2 increased from T0 to T2. It is not obvious to observe an increasing in plasma levels of IL-2 in patients after two cycles of exogenous IL-2 administration, because of its rapid clearance in humans [25]. Therefore, the increase of the IL-2 level might be due to an indirect effect mediated by the activation or expansion of immune cells able to produce IL-2, such as CD8+ T cells, CD4+ T cells and dendritic cells.

The increase of IL-5 observed at T2 could be associated to IL-2 administration. It has been shown that IL-5 increases after IL-2 therapy [26]. In other studies, IL-5 increased concomitantly with eosinophils [27,28] which were found after systemic [29] and locally IL-2 therapy [30]. We did not check eosinophil counts in our patient series, and we cannot confirm this observation. However, the increment in IL-5 levels should be regarded positive, since it could contribute to tumour control via eosinophil upregulation [31]. We also observed significantly higher levels of IL-8 at TPD in all patients, a result that supports the role of this cytokine in cancer progression and metastasis [32,33].

Our study is hampered by several weaknesses. First of all, IL-2 treatment in the era of immune checkpoint inhibitors seems anachronistic, but at the time of study design and approval, none of the modern immunotherapies were available for these types of tumour. Moreover, IL-2 has shown activity in many solid tumours [34], and the low IL-2 dose we used was not hampered by the heavy toxicities observed with higher doses [35] as confirmed also in the present experience. Second, this study is based on a limited number of patients and is exploratory in nature, due to the difficulties to identify end-stage patients that meet all the inclusion criteria. Lastly, we cannot speculate that the effects seen in the patients treated with this combination are similar to those achievable with other immunotherapies. Ongoing studies from our group, using a similar combination but replacing IL-2 with an anti-PD-L1 [36] will partially clarify this point. However, it is possible that different immune drugs, targeting other immune check-points, such as LAG-3, TIM-3 and VISTA, or other immune targets may induce different effects on the circulating cytokines.

## 5. Conclusions

Notwithstanding the above limitations, the data generated in our study suggest that the baseline circulating cytokines landscape in end-stage patients deserves further investigation. Cytokines could serve as better biomarkers than tumour type and other factors such as the NLR, number of previous treatment lines or LDH in selecting end-stage patients potentially suitable for IO. Our findings also suggest that the information on the landscape of circulating cytokines in end-stage cancer patients could provide a picture of the tumour–host interaction and therefore could aid in deciding which patients should be referred to palliative care. In conclusion, even considering the limitation of an exploratory study, our data support a central role of the immune system in driving outcome in cancer patients.

## Figures and Tables

**Figure 1 vaccines-09-00235-f001:**
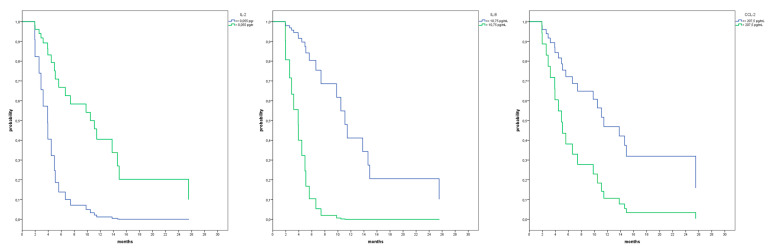
Cox analysis performed for IL-2 and 8 pts with an interleukin (IL)-2 concentration of ≤0.055 pg/mL and a 15 pts concentration of >0.055 pg/mL (hazard ratio (HR), 0.204; 95% CI, 0.075–0.559; *p* = 0.002). Cox analysis performed for IL-8 and 14 pts with an IL-8 concentration of ≤10.75 pg/mL and a 9 pts concentration of >10.75 pg/mL (HR, 0.096; 95%CI, 0.024–0.383; *p* = 0.001). Cox analysis performed for CCL-2 and 8pts with a CCL-2 concentration of ≤207.5 pg/mL and a 15 pts concentration of >207.5 pg/mL (HR, 0.338; 95% CI, 0.127–0.903; *p* = 0.031).

**Figure 2 vaccines-09-00235-f002:**
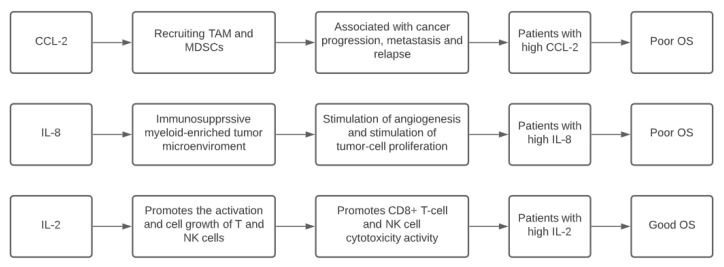
Most significant cytokines at T0, their functional meanings and effects of higher values on the patient’ outcome. MDSC: myeloid derived suppressor cells; TAM: tumour associated macrophages; NK: natural killer.

**Figure 3 vaccines-09-00235-f003:**
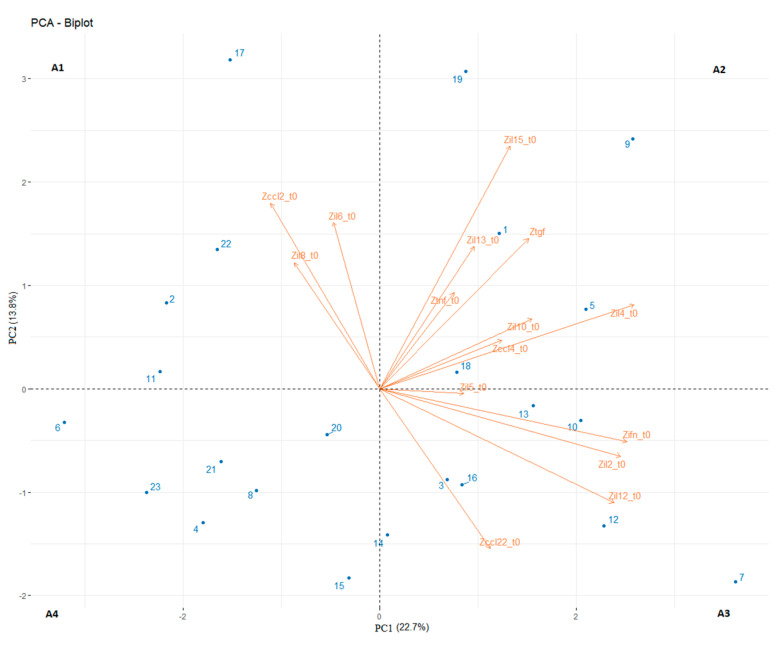
Principle component analysis (PCA) biplot of plasma cytokines (orange vectors) measured at baseline (T0) for all 23 patients (blue dots). In the x- and y-axes are plotted the two principal components (PC1 and PC2). Cytokines values are normalised in z-scores. Every number corresponds to a patient identified in Table 1. Area 1: PC1 < 0 and PC2 > 0; Area 2: PC1 > 0 and PC2 > 0; area 3: PC1 > 0 and PC2 < 0; area 4: PC1 < 0 and PC2 < 0.

**Figure 4 vaccines-09-00235-f004:**
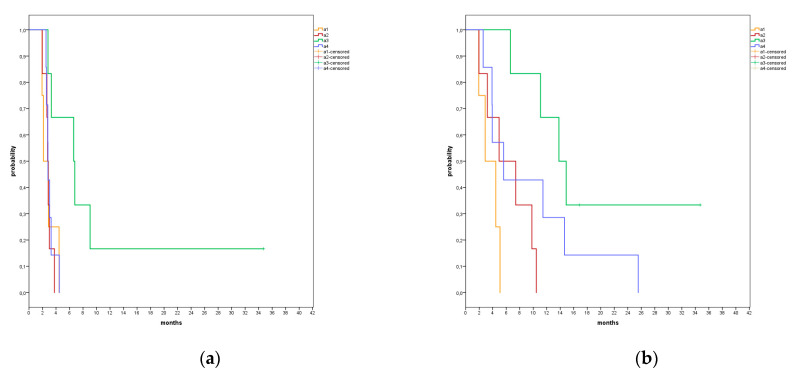
Kaplan–Meier test results for PFS (**a**) and for OS (**b**) divided in four areas (A1 (4 pts) = orange, A2 (5 pts) = red, A3 (7 pts) = green, A4 (7 pts) = blue).

**Figure 5 vaccines-09-00235-f005:**
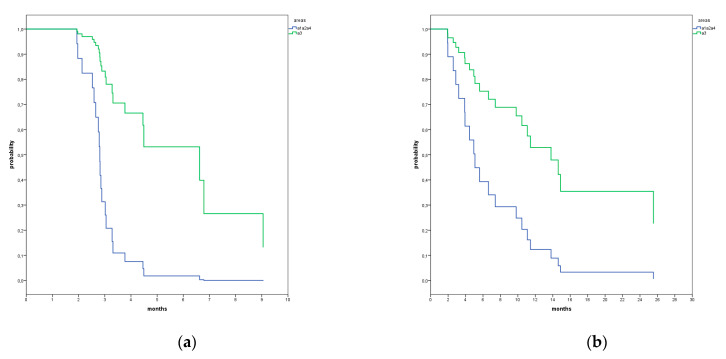
Cox analysis for PFS (HR: 0.158; 95% CI: 0.044–0.563; *p* = 0.004) (**a**) and for OS (HR: 0.304; 95% CI: 0.109–0.849) (**b**) showing the ensemble of areas 1, 2 and 4 (16 pts) in blue and in area A3 (7 pts) in green.

**Figure 6 vaccines-09-00235-f006:**
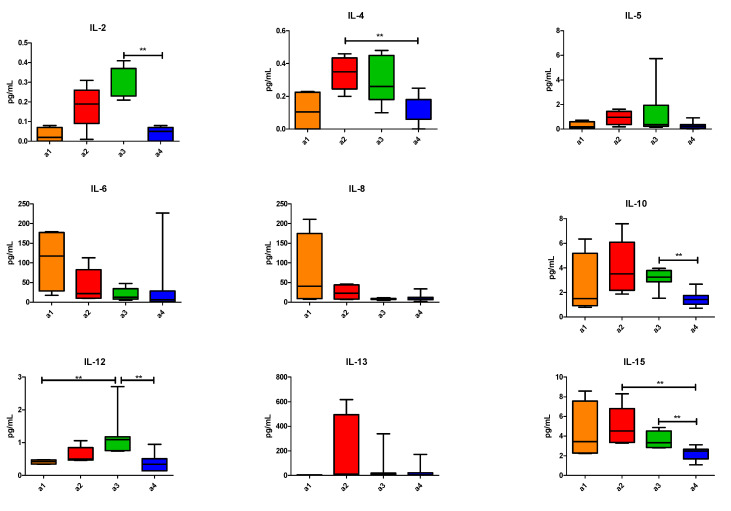
Distributions of cytokines among four areas: orange bars for area 1 (4 pts), red bars for area 2 (5 pts), green bars for area 3 (7 pts) and blue bars for area 4 (7 pts). Concentrations are express in pg/mL. Only *p* values corrected for multiplicity are shown. ** *p* ≤ 0.01.

**Figure 7 vaccines-09-00235-f007:**
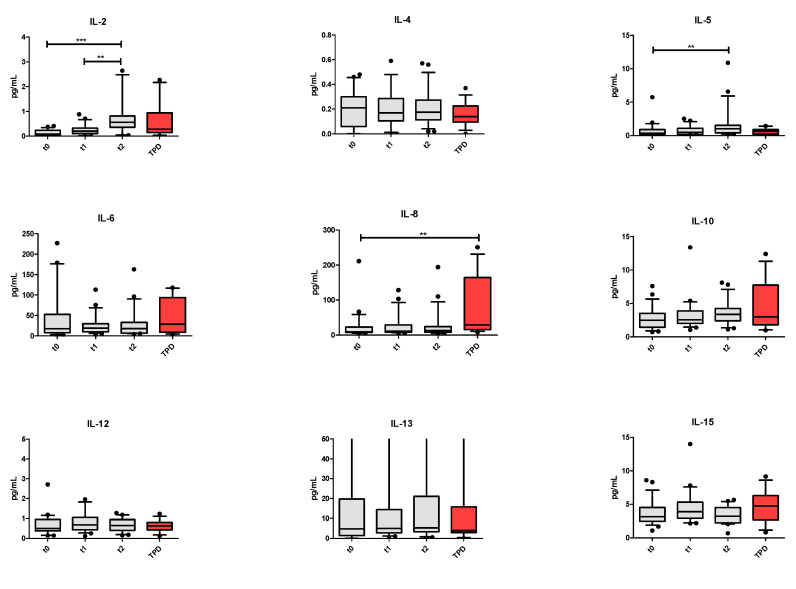
Distributions of cytokines among four time points: t0, t1, t2 (grey) and at disease progression (TPD; red). Only *p* values corrected for multiplicity are shown. ** *p* ≤ 0.01.

**Figure 8 vaccines-09-00235-f008:**
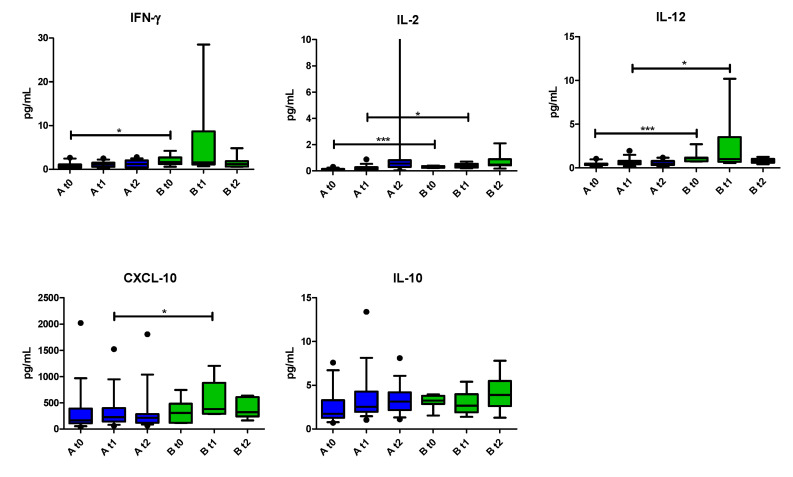
Distributions of five cytokines in group A (blue bars) and B (green bars). Only cytokines with any statistical significance or trend are shown. Concentrations are express in pg/mL. IL-2 for graphical purpose has a value (84.60 pg/mL) at t2 of group A that was intentionally removed by scaling the y-axis. * *p* ≤ 0.05; *** *p* ≤ 0.001.

**Figure 9 vaccines-09-00235-f009:**
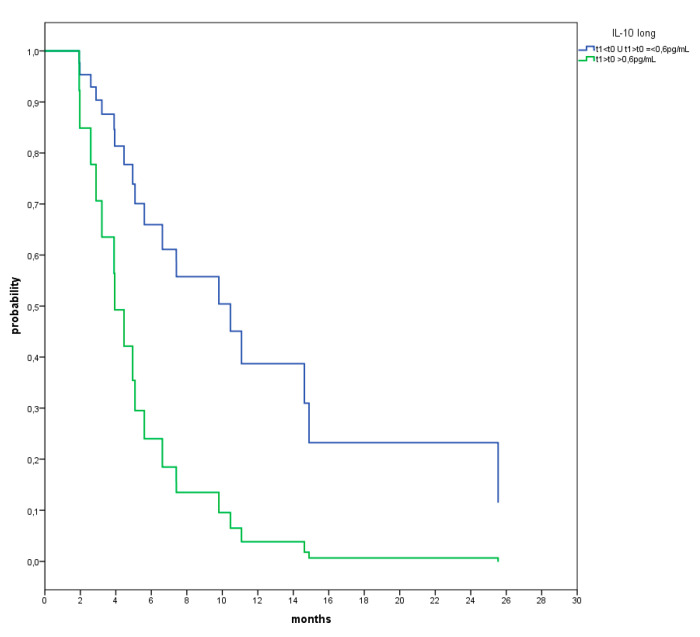
Cox analysis for OS between pts (blue line) which show IL-10 values at t1 lower than t0 (t1 < t0) or lower than a cut-off value of increment defined by ROC analysis (0.6 pg/mL) and pts (green line) which show an increment above the cut-off value (0.6 pg/mL) at t1 compared to at t0. (HR: 0.291; 95% CI: 0.103–0.822; *p* = 0.020).

**Figure 10 vaccines-09-00235-f010:**
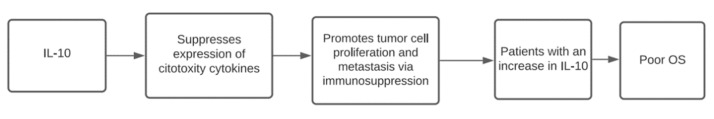
Changes of IL-10 from T0 to T1, its functional meanings, and its effect on the patients’ outcome.

**Table 1 vaccines-09-00235-t001:** Patients’ characteristics.

Patients	Age (Y.O.)	Sex	PS	N° of Prev. Th.	Types of Prev. Th.	Last But One Th-Best Response	Last Th-Best Response	TTF (Weeks)	NLR	LDH	OS (Weeks)	PCA
**COLON**												
TC1	50	F	1	3	Fol + Beva, Fil, Myt + Cape	PD	PD	19.6	5.6	511	111.1	15
TC2	72	F	1	5	Fol, Fil + Beva, Xil, Fil + 5FU, EDX	PD	PD	28.8	4.3	444	28.8	16
TC3	71	M	0	3	Xel + Beva, Fil + Beva, Pani	SD	PD	8.4	7.9	5942	8.4	17
TC4	69	F	0	5	Xel + Beva, Cape, Fil + Beva, Fol, Rego	SD	PD	12	5.6	707	45.6	18
TC5	48	F	1	3	Fil + Cetu, Fol + Beva, Cetu + CPT11	SD	PD	11.7	6.9	2142	21.6	19
TC6	64	M	0	3	Xel, Fil + Beva, Pani	PD	PD	13.6	7.9	566	63.7	20
TC7	76	M	0	6	Fol + Beva, Cape + Beva, Xil + Beva, Cape, Rego, Fil	PD	PD	9.4	5.1	842	17	21
TC8	73	M	1	2	Fil + Pani, Xol	SD	PD	9.3	9.3	4806	22.1	22
TC11	55	M	0	3	Fol, Fil + Beva, Xol	SD	PD	13.4	3.5	448	49.8	23
**KIDNEY**												
TK3	75	M	1	4	Sun, Eve, Sor, Cabo	PD	SD	39	7.0	338	73.3	13
TK2	68	M	0	4	Sun, Eve, Sor, NVB	PD	SD	177.6	2.4	485	177.6	12
TK1	49	M	1	5	Sun, Axi, Eve, Sor, Pazo	SD	PD	12.6	6.2	283	12.6	11
TK4	81	F	1	4	Sun, Pazo, NVB, Cabo	SD	SD	29.7	5.3	486	48.3	14
**PROSTATE**												
TP1	64	M	0	3	Doce, Caba, CTX	SD	SD	16.7	4.8	917	60.1	7
TP2	69	M	0	5	Doce, Caba, NVB, Enza, CTX	PR	PR	13.9	1.6	478	24.4	8
TP3	67	M	1	4	BAT, Doce, Caba, Enza	PR	PR	20	3.2	450	42.7	9
TP4	77	M	0	5	LHRHa, Doce, Abi, Caba, Enza	SD	PR	18.1	3.6	571	32.3	10
**BREAST**												
TB1	73	F	0	8	Exe, Cape + NVB + CTX, Ful, Doxo, NabPa, Cape, Eri, NVB	SD	PD	11.3	2.4	872	14	1
TB2	46	F	2	9	Tam, NVB, Cape, Meg, Ana, EDX + MTX, PegDoxo, Eri, NabPa	PD	PD	19.4	6.6	1431	19.4	2
TB3	62	F	1	4	Letro, Ful, Cape, CTX	PD	SD	9.9	4.2	821	454.1	3
TB4	64	F	0	8	FEC + Pacli, Letro, ExeFul, Cape, Eri, NVB, NabPa	PD	SD	13.3	3.4	621	17.1	4
TB5	79	F	1	8	Pacli, Ana, Exe, Ful, Cape, PegDoxo, Cape + NVB, Eri	PD	PD	8.6	2.2	787	8.6	5
TB8	66	F	2	8	FEC + Pacli, Letro, Eve + Exe, Ful, Cape + NVB, Eri, Doxo, NVB	SD	PD	11.3	5.0	378	11.3	6

Legend: PS, performance status; N° of prev. th., number of previous medical treatments; types of prev. th., types of previous medical treatments; last but one th.-best response, best response to the penultimate therapy; last th.-best response, best response to the last therapy; TTF, time to treatment failure; NLR, neutrofils/lymphocytes ratio; LDH, lactate dehydrogenase; OS, overall survival; PCA, principal component analysis; M, man; F, female; PD, progression disease; SD, stable disease; RP, partial response. Medical treatments: COLON: Fol, Folfox (folinic acid, fluorouracil (5FU) and oxaliplatin); Xil, Xeliri (irinotecan and capecitabine); Fil: Folfiri (folinic acid, fluorouracil and irinotecan); Xel: Xelox (oxaliplatine and capecitabine); EDX, epidoxorubicin; Cape, capecitabine; Fol + Beva, Folfox + bevacizumab; Fil + Beva, folfiri + bevacizumab; Xel+Beva, Xelox + bevacizumab; Cape + Beva, capecitabine + bevacizumab; Myt + Cape, mytomicin + capecitabine; Fil + 5FU, folfiri + De Gramont (5FU + folinic acid); Pan, panitumumab; Rego, regorafenib; Fil + Cetu, folfiri + cetuximab; Cetu + CPT11, cetuximab + irinotecan; KIDNEY: Sun, sunitinib; Eve, everolimus; Sor, sorafenib; Cabo, cabozantinib; Axi, axitinib; Pazo, pazopanib; NVB, vinorelbine; PROSTATE: Doce, docetaxel; Caba, cabazitaxel; CTX, cyclophosphamide; Enza, enzalutamide; Abi, abiraterone acetate; LHRHa, luteinizing hormone-releasing hormone (LHRH) analogues; BAT, total androgen blockage (bicalutamide + LHRHa); BREAST: Exe, exemestane; Tam, tamoxifene; Meg, megestrol; Ana, anastrozole; Letro, letrozole; Cape + NVB + CTX, capecitabine + vinorelbine + cyclophosphamide; Ful, fulvestrant; Doxo, doxorubicin; NabPa, nabpaclitaxel; Eri: eribulin; EDX + MTX, cyclophosphamide + metotrexate; PegDoxo, pegylated liposomal doxorubicin; FEC + pacli, 5FU, epirubicin, cyclophosphamide + paclitaxel.

**Table 2 vaccines-09-00235-t002:** Median Progression-free survival (PFS) and overall survival (OS) by the tumor primary site. mPFS, median PFS; mOS, median OS; CI, confidence interval; B, breast cancer; P, prostate cancer; K, renal cell carcinoma; C, colon cancer.

Group	mPFS (CI) (Months)	mOS (CI) (Months)
TB	2.7 (2.1–3.2)	3.2 (1.6–4.8)
TP	2.8 (2.6–3.0)	7.4 (3.3–11.5)
TK	6.8 (0.7–12.8)	11.1 (0.0–24.7)
TC	2.8 (2.6–3.0)	6.6 (2.1–11.1)

**Table 3 vaccines-09-00235-t003:** Univariate and multivariate Cox analysis. *N* = number of patients; HR = hazard ratio; S.E. = standard error; CI = confidence interval; N. previous Th. = number of previous medical treatments.

Univariate Cox Model for PFS						
Variables	Group	*N*	HR	S.E.	95% CI	*p* Value
Cancer site	Kidney	4	0.093	1.049	0.012–0.723	0.023
	Other sites	19	1			
N° prev. Th.	≤4 prev. Th.	12	0.549	0.456	0.225–1.343	0.189
	>4 prev. Th.	11	1			
Areas	Area 3	7	0.158	0.650	0.044–0.563	0.004
	Others	16	1			
**Multivariate Cox Model for PFS**						
Cancer site	Kidney	4	0.178	1.115	0.020–1.587	0.122
	Other sites	19	1			
N° prev. Th.	≤4 prev. Th.	12	0.796	0.467	0.318–1.988	0.625
	>4 prev. Th.	11	1			
Areas	Area 3	7	0.261	0.666	0.071–0.962	0.044
	Others	16	1			
**Univariate Cox Model for OS**						
Cancer site	Kidney	4	0.273	0.754	0.062–1.194	0.085
	Other sites	19	1			
N° prev. Th.	≤4 prev. Th.	12	0.148	0.613	0.044–0.491	0.002
	>4 prev. Th.	11	1			
Areas	Area 3	7	0.304	0.524	0.109–0.849	0.023
	Others	16	1			
**Multivariate Cox Model for OS**						
Cancer site	Kidney	4	0.712	0.875	0.128–3.960	0.698
	Other sites	19	1			
N° prev. Th.	≤4 prev. Th.	12	0.177	0.612	0.053–0.588	0.005
	>4 prev. Th.	11	1			
Areas	Area 3	7	0.396	0.599	0.122–1.282	0.122
	Others	16	1			

## Data Availability

Data supporting reported results can be found at the ARCO foundation laboratory in the Santa Croce e Carle Teaching Hospital (Cuneo, Italy).

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
