# Peer review of "Cytokine Profiling of End Stage Cancer Patients Treated with Immunotherapy"

_vaccines, 2021, doi:10.3390/vaccines9030235_

Round 1

Reviewer 1 Report

The manuscript of Merlano et al.  studied the cytokine profiling at the end stage cancer patients treated with immunotherapy. The authors used the following therapy regimen: metronomic cyclophosphamide+ low dose IL-2+ a single radiation shot , then they measured the cytokine panels before treatment , after RT (T1), at cycle 2 (T2), and at disease progression. They studied the cytokines in 23 patients  including breast cancer, prostate cancer, kidney cancer, and colon cancer. The authors reported that high IL-2, low IL-8 and CCL-2 correlated with overall survival. In addition, higher IL-8 correlated to the disease progression, while higher IL-10 negatively correlated to overall survival. The author concluded that the baseline cytokine levels could be a predicative biomarker for the patients’ outcome and affect the the therapy kinetics at the end stage of cancer.

Overall, the manuscript is nicely written, well designed, and has an acceptable flow.

I have minor comments

1- The authors measured 16 cytokines at different time points. Why the authors choose these cytokines for analysis? Did the authors try proteome profile array?

2- In the statistic section, the authors used non-parametric Mann Whitney U test, which is non paired test. Since the authors measured cytokines in the same cohort at different time point, it is better to analysis the results using paired t-test. 

Author Response

1- The authors measured 16 cytokines at different time points. Why

the authors choose these cytokines for analysis? Did the authors

try proteome profile array?

We partially answered this question in the original manuscript (section 2.4, lines 109-110).

Now we have described in the “Introduction” section why we have chosen these cytokines for analysis (Introductions, lines 68 – 70). We hope this change may clarify this point.

We did not try proteome profile array.

2- In the statistic section, the authors used non-parametric Mann

Whitney U test, which is non paired test. Since the authors

measured cytokines in the same cohort at different time point, it is

better to analysis the results using paired t-test.

It is true that the series would be suitable to be analyzed with paired t-test, but nevertheless several outliers were present in our cohort and variables analyzed do not follow a normal distribution, therefore we preferred to compute a comparison between medians values.

1- The authors measured 16 cytokines at different time points. Why

the authors choose these cytokines for analysis? Did the authors

try proteome profile array?

We partially answered this question in the original manuscript (section 2.4, lines 109-110).

Now we have described in the “Introduction” section why we have chosen these cytokines for analysis (Introductions, lines 68 – 70). We hope this change may clarify this point.

We did not try proteome profile array.

2- In the statistic section, the authors used non-parametric Mann

Whitney U test, which is non paired test. Since the authors

measured cytokines in the same cohort at different time point, it is

better to analysis the results using paired t-test.

It is true that the series would be suitable to be analyzed with paired t-test, but nevertheless several outliers were present in our cohort and variables analyzed do not follow a normal distribution, therefore we preferred to compute a comparison between medians values.

1- The authors measured 16 cytokines at different time points. Why

the authors choose these cytokines for analysis? Did the authors

try proteome profile array?

We partially answered this question in the original manuscript (section 2.4, lines 109-110).

Now we have described in the “Introduction” section why we have chosen these cytokines for analysis (Introductions, lines 68 – 70). We hope this change may clarify this point.

We did not try proteome profile array.

2- In the statistic section, the authors used non-parametric Mann

Whitney U test, which is non paired test. Since the authors

measured cytokines in the same cohort at different time point, it is

better to analysis the results using paired t-test.

It is true that the series would be suitable to be analyzed with paired t-test, but nevertheless several outliers were present in our cohort and variables analyzed do not follow a normal distribution, therefore we preferred to compute a comparison between medians values.

Reviewer 2 Report

This is a very well conducted study which adds a good value to the current knowledge and provides excellent set of data which is of interest for a broad audience.

I would however, suggest that the Authors need to perform indicated revisions in order for the work to be accepted for publication. 

1) The introduction is too short and does not explain the purpose of the work properly. I would suggest that the Authors re-work and expand it and explain the roles of the growth factors and cytokines they measure and provide an explanation of why these particular proteins were quantified. 

Minor point - why abbreviating "Immunotherapy" as "IO" not "IT"?

2) The Authors use the term "cytokines" to identify the proteins they measure. But transforming growth factor beta (TGF-beta) is not a cytokine  - it is growth factor. Sometimes in the literature "growth factors" are called "cytokines", which is biochemically incorrect. Growth factors can only trigger "positive" responses, while cytokines (e. g. TNF-alpha) can induce programmed cell death. 

3) In Methodology section - a brief description of Simple Plex System used to measure cytokines/growth factors would be beneficial. Also description of choices of factors to measure needs to be provided in the Introduction and in more details and not in Methodology section.

4) The Authors just provide results of cytokine/growth factor measurements. However, there is no summary of scientific meaning of the results. The paper would benefit from a table (or a scheme/diagram), where it is summarised the following - cytokine/growth factor measures, effect observed and then - functional meaning of the effect. Since this paper is of interest for broad audience, such a table or diagram would be really helpful. 

5) Both Introduction and Discussion. In the Introduction, the Authors briefly mention PD-1 and CTLA4 as targets for the immunotherapy. These are the most well-studied targets. But cancer immune evasion pathways appear to be more complex than just these two checkpoints. Firstly these are LAG-3, Tim-3/galectin-9 pathways, VISTA (PD-1 homologue or B7-homologue 5), galectin-9 which play major roles and there is often a cross-talk  between biochemical pathways associated with these checkpoint proteins. I would suggest that the Authors expand on this in the Introduction and also comment on this in the Discussion when interpreting their results. 

Author Response

1) The introduction is too short and does not explain the purpose of

the work properly. I would suggest that the Authors re-work and

expand it and explain the roles of the growth factors and cytokines

they measure and provide an explanation of why these particular

proteins were quantified.

We added a statement (lines 68 – 71) to clarify the purpose of the work.

We completed the “introduction” section with the list of the proteins analyzed with their functions (that explains why they are quantified): Introduction, lines 72 – 75. We also added statements on new immunotherapy agents under development for partially answer to question 5: introduction, lines 47 - 52.

Minor point - why abbreviating "Immunotherapy" as "IO" not "IT"?

"IT" and "IO" are both acronyms used in immuno-oncology articles, although "IT" is also used to identify "information technology". We prefer to use "IO" because it is more used in clinical oncology articles dedicated to immunotherapy.

2) The Authors use the term "cytokines" to identify the proteins

they measure. But transforming growth factor beta (TGF-beta) is

not a cytokine - it is growth factor. Sometimes in the literature

"growth factors" are called "cytokines", which is biochemically

incorrect. Growth factors can only trigger "positive" responses,

while cytokines (e. g. TNF-alpha) can induce programmed cell

death.

The reviewer is absolutely right. However, we use “cytokines” for brevity.

Indeed, in our opinion, detailing “interleukins, chemokines and growth factors” when we write about the panel of 16 proteins, may became boring for the readers.

3) In Methodology section - a brief description of Simple Plex

System used to measure cytokines/growth factors would be

beneficial. Also description of choices of factors to measure needs

to be provided in the Introduction and in more details and not in

Methodology section.

Following the Reviewer’s suggestion, we added two sentences (see text lines 128-130) to simply describe the Simple Plex System.

We have moved the description of the factors to measure in the introduction (see text and the answer to question 1)

4) The Authors just provide results of cytokine/growth factor

measurements. However, there is no summary of scientific

meaning of the results. The paper would benefit from a table (or a

scheme/diagram), where it is summarised the following -

cytokine/growth factor measures, effect observed and then -

functional meaning of the effect. Since this paper is of interest for

broad audience, such a table or diagram would be really helpful.

We appreciated the suggestion. Therefore, we added two figures (figure 2 and figure 10) where we summarized the effects of the most relevant cytokines observed at T0, their functional meaning and their effects (figure 2) and the effects of longitudinal changes of IL-10, the only whose changes are significantly correlated with outcome (figure 10). To link them with text, we added two short sentences (line 236 and line 336).

5) Both Introduction and Discussion. In the Introduction, the

Authors briefly mention PD-1 and CTLA4 as targets for the

immunotherapy. These are the most well-studied targets. But

cancer immune evasion pathways appear to be more complex than

just these two checkpoints. Firstly these are LAG-3, Tim-3/galectin-

9 pathways, VISTA (PD-1 homologue or B7-homologue 5),

galectin-9 which play major roles and there is often a cross-talk

between biochemical pathways associated with these checkpoint

proteins. I would suggest that the Authors expand on this in the

Introduction and also comment on this in the Discussion when

interpreting their results.

We have followed the Review’s suggestions and modified both introduction and discussion accordingly. In details, changes in the “introduction” are detailed in the answer to the query # 1; changes in the “discussion” have been added from line 427 to line 433.
